# Attenuating Mutations in Usutu Virus: Towards Understanding Orthoflavivirus Virulence Determinants and Live Attenuated Vaccine Design

**DOI:** 10.3390/vaccines13050495

**Published:** 2025-05-03

**Authors:** Johanna M. Duyvestyn, Peter J. Bredenbeek, Marie J. Gruters, Ali Tas, Tessa Nelemans, Marjolein Kikkert, Martijn J. van Hemert

**Affiliations:** Molecular Virology Laboratory, Leiden University Center for Infectious Diseases (LUCID), Leiden University Medical Center, 2333 ZA Leiden, The Netherlands; j.m.duyvestyn@lumc.nl (J.M.D.); m.j.van_hemert@lumc.nl (M.J.v.H.)

**Keywords:** orthoflavivirus, Usutu virus, vaccine, attenuation, animal model, emerging virus, arbovirus

## Abstract

Background/Objectives: Understanding virulence determinants can inform safer and more efficacious live attenuated vaccine design. However, applying this knowledge across related viruses does not always result in conserved phenotypes from similar mutants. Methods: Using Usutu virus (USUV), an emerging orthoflavivirus spreading through Europe, we assessed whether the attenuating effect of the mutations described for related orthoflaviviruses is conserved. Candidate attenuating mutations were selected based on previous studies in other orthoflaviviruses and incorporated into USUV. Results: Nine variants, with mutations in the USUV envelope, non-structural (NS) proteins NS1, NS2A, or NS4B were stable and selected for further characterisation. The variants with an attenuating phenotype in cell culture were then compared to the wild-type virus in an Ifnar^−/−^ mouse model. Mutations of the envelope glycosylation sites and glycosaminoglycan binding sites, which were recognised as more-conserved mechanisms of orthoflavivirus attenuation, were attenuating in USUV as well. However, not all the mutations explored in the USUV non-structural proteins exhibited an attenuated phenotype. Instead, the attenuation was either less pronounced, or there was no change in phenotype relative to the wild-type virus at all. Conclusions: In addition to improving our understanding of USUV virulence determinants, these results add to a growing body of literature highlighting the most promising mechanisms to target for the design of safe live attenuated vaccines against emerging orthoflaviviruses.

## 1. Introduction

The *Orthoflavivirus* genus, family *Flaviviridae*, includes arthropod-borne single-stranded (+)RNA viruses with a heavy disease burden in both animals and humans. The past decades have seen an increase in the global distribution for endemic, emerging, and reemerging orthoflaviviruses [1]. While some of these viruses are quite well characterised, such as dengue virus (DENV) or yellow fever virus (YFV), many others are less well understood. For emerging viruses like Usutu virus (USUV), relatively little is known. USUV is a member of the Japanese Encephalitis complex serogroup of orthoflaviviruses and is maintained in a zoonotic cycle between avian hosts and Culex species mosquitoes. In recent decades, USUV has emerged from Africa into Europe, where it is predominantly found in blackbirds, in which it has caused massive die-offs [2,3]. In humans, USUV infection is usually asymptomatic, but in rare cases it can manifest as visceral or neurological disease [4]. The prevention of future outbreaks and a proper response to this emerging orthoflavivirus require more research into the fundamental biology of these zoonotic viruses, their hosts, and their vector distribution [5].

Vaccines are one important countermeasure against orthoflavivirus outbreaks. Licensed vaccines have been developed against YFV, Japanese encephalitis virus (JEV), tick-borne encephalitis virus (TBEV), and DENV, with varying levels of success. However, for many orthoflaviviruses, no vaccines are available [6,7]. Some vaccine design strategies, including inactivated vaccines and mRNA formulations, are promising but present challenges in achieving a long-term immune response. Live attenuated vaccines on the other hand can provide life-long protection, as exemplified by the highly effective yellow fever 17D vaccine. However, the use of an actively replicating virus confers risks of adverse events, such as reversion to virulence or problems with clearance in immunosuppressed vaccinees [6,8]. In the past, such attenuated vaccine candidates have generally been developed by serial passaging, which can be regarded as a “black-box approach” as the precise effects of the resulting mutations usually remain unclear. However, advances in reverse genetics systems and the subsequent improved understanding of virulence determinants have opened avenues for the development of safe-by-design live attenuated vaccines [6,9].

Safe-by-design live attenuated vaccine strategies target specific determinants of virulence to create rationally attenuated vaccine candidates that retain an efficacious immune response but do not cause disease [9,10]. This requires finding a balance between replication levels and pathogenicity levels. These strategies require an in-depth understanding of the virus in question, including the function of the specific proteins or motifs that are involved in virulence. However, for emerging viruses, the knowledge of specific virulence determinants is likely limited, other than what may be inferred from closely related family members [7]. Numerous attenuation strategies have been assessed for the more well-known orthoflaviviruses. However, whether mutations that attenuate one specific orthoflavivirus also have a similar effect on other family members is often not clear [7,10]. Some mechanisms of attenuation appear to be broadly conserved across the orthoflavivirus genus, others are likely to be more serotype- or virus-specific, and many have only been studied in a single virus isolate (reviewed in van Bree et al. 2023 [10]). An improved understanding of rationally designed attenuating mutations and their conservation across a range of viruses could therefore aid in the development of live attenuated vaccine candidates against future (re)emerging viruses.

Mutations that influence viral glycosaminoglycan (GAG) binding have been demonstrated to be attenuating across a broad range of both orthoflaviviruses and alphaviruses [10,11,12,13,14,15,16,17,18,19]. GAGs are negatively charged polysaccharides that are expressed on the surface of mammalian cells and can serve as attachment factors for orthoflavivirus entry [20]. Mutations that increase the positive charge of certain regions of the viral envelope proteins result in the enhanced binding of viruses to these polysaccharides. Viruses with this phenotype often show enhanced growth in cell culture. However, in vivo, the enhanced GAG-binding phenotype results in the rapid clearance of the virus from the blood, reduced titres in peripheral infection, and loss of neuroinvasiveness—traits that decrease the pathogenicity of the virus. Attenuation in an animal model has been observed for numerous cell-culture-adapted viruses in which an increased GAG-binding phenotype has arisen, for example, in variants of JEV [15,19], YFV [16], DENV [14], TBEV [13,17], Murray Valley encephalitis virus (MVEV) [15], West Nile virus (WNV) [19], and Tembusu virus (TMUV) [12].

Another well-conserved mechanism of attenuation in orthoflaviviruses is the engineered removal of the glycosylation sites in the NS1 protein. NS1 is a complex non-structural glycoprotein that forms both dimers and hexamers, facilitating its numerous different intracellular and extracellular roles in replication and pathogenesis [21]. The loss of one or more of the highly conserved N-glycosylation sites disrupts the maturation and oligomerisation of the protein, interfering with NS1’s role in replication and its ability to form secreted hexamers—structures that have been implicated strongly in viral pathogenesis [22]. In vivo studies on DENV, YFV, WNV, ZIKV, and TMUV have all shown that mutating (or removing) the NS1 glycosylation sites results in significant increases in survival after infection [10,23,24,25,26,27].

Many other mutations within the non-structural proteins are attenuating in certain orthoflaviviruses, but whether these effects are conserved in other orthoflaviviruses is not yet known. For example, a single amino acid mutation, A30P, in NS2A (a transmembrane protein implicated in viral replication and innate immune evasion roles) reduces the lethality of the Kunjin (KUNV) strain of WNV in mouse models [28]. This mutation disrupts the formation of NS1’, a protein specific to the JEV serocomplex, which results from an extension of the NS1 open reading frame with 52 amino acids due to a slippery sequence in the mRNA secondary structure, causing a ribosomal frame shift [29]. The role of NS1’ in JEV complex viruses is not well understood. It may be used as a strategy simply to produce relatively more NS1 without translating the rest of the polyprotein, or it might play a role in avian pathogenesis or the central nervous system tropism of the JEV serocomplex orthoflaviviruses [29,30,31]. The A30P mutation also appears to at least partially disrupt the ability of KUNV NS2A to antagonise the innate immune response [28]. The effect of the A30P mutation has also been studied in WNV and JEV, with mixed results. In two different WNV strains, the attenuated phenotype was either absent or milder than observed for KUNV [32,33]. The serially passaged SA14-14-2 live attenuated JEV vaccine contains a mutation at NS2A-V23I, which also abolishes the production of NS1’. When introduced into the virulent SA14 JEV strain by reverse genetics, the virus became attenuated in a mouse model [34]. In contrast, however, in two other JEV strains, the same mutation abolished the expression of the NS1’ protein but did not confer an attenuated phenotype [35].

A number of mutations in the transmembrane protein NS4B have been reported for which the attenuating phenotype could be conserved across orthoflaviviruses, three of which will be discussed here [36,37,38,39]. First, the N-terminal region of NS4B contains conserved amino acids that could contribute to a role in innate immune antagonism [38]. The mutation of a highly conserved proline (P) residue in this region to a glycine (but not an alanine) resulted in a strongly attenuated WNV infection in animal models [37]. The corresponding P36A mutation in ZIKV resulted in a mildly attenuated phenotype in mice [38]. Second, a number of attenuating mutations in NS4B, reported in studies on WNV, ZIKV, DENV, JEV and YFV, map to a central hydrophobic region of the protein [37,38]. Many of these have only been assessed in a single member of the above-mentioned orthoflaviviruses. However, a C102S mutation in WNV and the corresponding C100S mutation in ZIKV, in a highly conserved site, were strongly attenuating in animal models for both viruses [37,38,39] (Appendix A). Thirdly, a substitution in the C-terminal tail of NS4B (E249G) was found to be conserved in naturally attenuated variants of WNV [40,41]. The mutation targets a conserved negatively charged residue within the JEV serocomplex orthoflaviviruses [40] (Appendix A), but the mechanism of this attenuation is not well understood. When introduced into a WNV replicon, the NS4B-E249G mutation decreased RNA synthesis [40], although, when introduced into a pathogenic WNV strain, this mutation was not attenuating in vivo [32].

In the current study, we introduced a variety of mutations in USUV, informed by the results from the above-mentioned studies, to assess whether the attenuating effect is also observed for USUV. After assessing the stability of the mutations over several passages in cell culture, we determined whether the expected phenotype was observed in vitro before testing whether these mutations were attenuating in a mouse model of disease. By incorporating these mutations into an emerging and relatively poorly studied orthoflavivirus, we aimed to gain a better understanding of the potential of these sites to be included in future vaccine development strategies.

## 2. Methods

### 2.1. Viruses

The wild-type USUV stock used was Africa-3 Lineage isolate AS201600045 TM Netherlands 2016 (Af-3-NL), GenBank accession number MH891847 [42]. The stock was received from Erasmus Medical Center Rotterdam, The Netherlands, and was passaged twice on Vero CCL-81 cells at 37 °C, 5% CO_2_ in Viral Growth Media made up of Dulbecco’s modified Eagle’s medium (DMEM, Gibco, Thermo Fisher (Life technologies), Landsmeer, The Netherlands) supplemented with 8% foetal calf serum (FCS, Capricorn Scientific, Ebsdorfergrund, Germany), 100 units/mL of streptomycin/penicillin (Sigma-Aldrich, Zwijndrecht, The Netherlands), 1% sodium bicarbonate (NaHCO_3_, Gibco), and 2 mM L-glutamine (Sigma-Aldrich). Infectious virus titre was determined by plaque assay on BHK-21J cells.

### 2.2. Recombinant USUV cDNA Clones

A wild-type USUV Africa-3 lineage cDNA clone (rUSUV-WT) was built following a TAR recombineering protocol in yeast, adapted from the method described in Thi Nhu Thao et al., 2020 (Genbank Accession PQ041659.1) [43,44]. Briefly, overlapping fragments of the USUV genome were amplified by PCR using cDNA from Af-3-NL as a template. The purified PCR products were assembled into the pCC1BAC-his3 vector by transformation-associated recombination (TAR) in *S. cerevisiae*. Purified DNA was isolated from colonies screened for correct assembly and transformed into *E. coli* for large-scale plasmid extraction. The plasmid was linearised and reverse-transcribed, and then the purified RNA was electroporated into BHK21-J cells as described previously [45]. After 4 days, the supernatant was harvested from the cells and used to inoculate Vero CCL-81 cells in order to obtain a p1 stock. The full genome was sequenced by next-generation sequencing (NGS) to confirm that no mutations had arisen during the construction and passaging of the clone.

The mutant viruses (E-E138K, E-E306K, NS1-AAA, NS1-QQAAA, NS2A-A30P, NS4B-P41G, NS4B-C105S, NS4B-D252G) were generated following this same TAR recombineering protocol using the above recombinant DNA clone as a template. The fragment containing the region of interest for each respective mutation was cloned into the pCR™8/GW/TOPO vector (Thermo Fisher), and the desired nucleotide changes were made by site-directed mutagenesis (Primers in Appendix A). The generated PCR products (with mutations) were then purified, and the TAR recombineering protocol was followed to assemble the mutant plasmids and launch the mutant viruses. The full genomes of the bacmids were checked by Sanger sequencing, and the RNA isolated from the passage 4 virus stock was sequenced by NGS to confirm the presence of the mutation and absence of other (undesired) mutations.

### 2.3. Cell Lines

Vero CCL-81 cells (LUMC cell line collection identifier VeroMM-2) and A549 cells were both cultured in DMEM supplemented with 8% FCS and 100 units/mL of streptomycin/penicillin. BHK21-J cells [46] were cultured in Glasgow’s MEM (GMEM, Gibco) supplemented with 8% FCS, 10% tryptose phosphate broth (Gibco), 10 mM HEPES (Lonza, Verviers, Belgium), and 100 units/mL of streptomycin/penicillin. All cells were maintained at 37 °C in a 5% CO_2_ incubator.

### 2.4. Viral Growth Kinetics

Cells were grown to 80% confluency in multi-well plates. The medium was removed, and the cells were infected at an MOI of 0.01 (Vero CCL-81, BHK21-J cells) or MOI of 1 (A549 cells) for 1 h at 37 °C. After removal of the inoculum, cells were washed gently three times with PBS before adding viral growth medium. After incubation at 37 °C, the supernatant was collected at the specified time points. Triplicate experiments were performed, and a representative replicate is depicted in the figures.

### 2.5. Virus Quantification

Reverse-transcription quantitative PCR (RT-qPCR), plaque assays, and TCID50 assays were performed as previously published [44]. In brief, absolute viral RNA copy numbers were determined by internally controlled multiplex TaqMan RT-qPCR on isolated RNA samples. The primers used are listed in Appendix A. Plaque assays were performed on BHK21-J cells and counted after 4 days of incubation. The detection limit was 20 pfu/mL in the 6-wells format or 40 pfu/mL in the 2-wells format. Plaque size was measured as the diameter of an approximated circle of the plaque, and an average of this was taken for all plaques in a single well. Infectious titre on Vero CCL-81 cells was determined by TCID50 assay by scoring cytopathic effect (CPE) at day 5 after infection. Vero CCL-81 monolayers infected with USUV did not develop clear or reliable plaques. The detection limit of this assay was 31.6 TCID50/mL. Virus-infected cells were incubated at 37 °C or at 41 °C (and counted one day earlier) for temperature sensitivity assays.

### 2.6. Western Blots

Cells infected with mutant or wild-type viruses and grown for 24 or 48 h were lysed in 2× Laemmli sample buffer. Proteins were separated by sodium dodecyl sulphate-polyacrylamide gel electrophoresis (SDS-PAGE) and then transferred onto a nitrocellulose membrane by semi-dry blotting using a Trans-blot Turbo system, version 1.02 (Bio-rad, Veenendaal, The Netherlands). Membranes were blocked in 5% dried milk powder in PBS with 0.05% Tween-20 (PBST) for 1 h, followed by overnight incubation with mouse anti-NS1 D/2/D6/B7 antibody (Abcam, Cambridge, UK), and diluted 1:1000 in PBST + 5% bovine serum albumin (BSA) at 4 °C. After incubation with a horseradish-peroxidase-conjugated secondary antibody, the blots were visualised with Clarity Western ECL substrate (Bio-rad)). Raw data for Western blots are shown in Appendix A.

### 2.7. Heparin Inhibition Assay

Heparin (Sigma-Aldrich) was diluted in infection media in a 1:2 serial dilution series from 160 μg/mL to 2.5 μg/mL. BHK21-J cells were grown to 70% confluency and preincubated with the heparin dilutions or media only (untreated control) for 15 min at room temperature. The preincubation media on the cells was replaced with 100 pfu/well of virus inoculum in triplicate for rUSUV-WT, E-E138K, or E-306K or with diluent only (DMEM with 3% FCS and 100 units/mL streptomycin/penicillin). The wells were incubated at 37 °C for 1 h; the media were replaced with overlay media and then incubated for 4 days before plaques were counted. Three replicate experiments using heparin concentrations of 10, 100, and 200 μg/mL confirmed these results.

### 2.8. Mouse Studies

Ifnar^−/−^ mice in a C57BL/6 background (B6(Cg)-Ifnar1<tm1.2Ees>/J) were purchased from The Jackson Laboratory (Bar Harbor, ME, USA) and subsequently bred and maintained in pathogen-free facilities at the LUMC Central Animal Facility (PDC) at 20–22 °C, a humidity of 45–65% RV, and a light cycle of 6:30–7:00 h sunrise, 07:00–18:00 h daytime, and 18:00–18:30 h sunset. Mice had access to water and food ad libitum and were provided with cage enrichment. Age-matched male and female mice were arranged in groups of n = 8 (virus groups) or n = 3 or 8 (controls) and acclimated to the experimental facility for 7 days. Mice were inoculated with 100 µL of 1200 TCID50 of virus (or 100 pfu in the case of the P41G mutant) in DMEM or DMEM alone (mock infection) via sub-cutaneous (SC) injection into the hind flank. Plaque assays on BHK21-J cells and/or TCID50 assay on Vero CCL-81 cells were performed with the virus inocula to confirm that the mice were inoculated with the intended dose for each animal experiment (Appendix A).

Mice were weighed and monitored for health and comfort daily, with special attention for the following aspects: activity, coat condition, hind limb function, and ocular discharge. Sera from tail vein bleeds were collected on alternating days. Upon reaching defined humane endpoints or at the end of the experiment, mice were euthanised with CO_2_, and a final serum sample was taken by heart puncture. Surviving animals and mock-infected mice were harvested at day 14 or at least 5 days after the last infected animal succumbed. Tissue samples from liver, spleen, kidney, and brain were dissected, weighed, and placed in viral transport medium (VTM: MEM without L-glut, HEPES-buffered, 100 units/mL of streptomycin/penicillin, amphotericin B, gentamycin, and 10% glycerol) and stored at −80 °C. For virus quantification, tissues in VTM were homogenised by pulsation with mixed sizes of acid-washed glass beads, 425–600 μm (Sigma-Aldrich) and 3 mm (VWR International, Amsterdam, The Netherlands), in a PRECELLYS^®^ 24 Tissue homogeniser (Bertin Instruments, Montigny-le-Bretonneu, France), then centrifuged to obtain a clear supernatant. RNA was isolated, viral load was determined by RT-qPCR, and infectious titres of RT-qPCR-positive samples were determined by plaque assay or TCID50 assay. Serum samples from mouse bleeds were inactivated with 0.2% Triton-X, diluted 1:20 and used directly for RT-qPCR.

### 2.9. Sequencing of Animal Samples

NGS was used to obtain the USUV genome sequences from the mouse brains at the end of the in vivo experiments. RNA was isolated from the homogenised tissue samples using the Bio-on-Magnetic-Beads (BOMB) method [47] for Illumina sequencing at GenomeScan B.V. (Leiden, The Netherlands). Sanger sequencing was used to assess the presence of the designed mutation in the USUV genomes isolated from mouse sera. Inactivated sera samples from the timepoint showing peak viremia (day 6–8 depending on the mutant virus) were used for reverse-transcription of RNA into cDNA using RevertAid H minus reverse-transcriptase (Thermo Fisher Scientific) and USUV-specific primers (Appendix A). The PCR-amplified product was then analysed by Sanger sequencing.

### 2.10. Virus Neutralisation Assay

Heat-inactivated (30 min at 56 °C) serum samples were serially diluted three-fold in DMEM medium supplemented with streptomycin/penicillin and 3% FCS, with an initial dilution of 1:10 and a final dilution of 1:21870. Diluted sera were mixed with equal volumes of 100 TCID50/mL USUV Af-3-NL and incubated for 1 h at 37 °C. The virus–serum mixtures were then transferred onto semiconfluent Vero CCL-81 cell monolayers and incubated at 37 °C and 5% CO_2_. Cells with media only or 100 TCID50/mL USUV Af-3-NL were used as negative (uninfected) and positive (infected) controls, respectively. Mock sera were pooled from mock mice over two replicate experiments, and positive control sera were pooled from mice infected with USUV Af-3 wild-type virus. At 5 days after infection, the cytopathic effect was evaluated (as positive or negative). The virus neutralisation titre is expressed as the reciprocal value of the highest dilution of the serum that still inhibited virus replication. An USUV Af-3-NL b-ack titration was included with each assay run to confirm the dose of the inoculum.

### 2.11. Statistical Analysis

Statistical analyses were performed in GraphPad Prism (version 9). All data are presented as mean ± SEM, unless stated otherwise. Survival experiments were analysed using the log-rank (Mantel–Cox) test. Viral titres in mouse sera were analysed using one-way ANOVA for each timepoint. Viral titres of mouse tissues were analysed using unpaired *t*-test corrected for multiple analyses.

## 3. Results

### 3.1. Design of Potentially Attenuated USUV Mutants

We selected the mutations based upon the amino acid changes that were attenuating in USUV-related viruses within the Japanese Encephalitis complex or that target mechanisms that are known to be broadly conserved within the orthoflaviviruses [7,10]. All the mutations investigated in this study are presented schematically in Figure 1.

We designed two mutations predicted to enhance GAG binding, E-E138K, and E-E306K (Figure 1), identified as part of the mechanism of attenuation of the serially passaged live attenuated JEV SA-14-14-2 vaccine [12,15]. Attenuation resulting from increased GAG binding has been well characterised in the cell-adapted variants of a number of different orthoflaviviruses [10], but whether this phenotype can be obtained by rational design, i.e., by introducing specific mutations into the envelope protein of related orthoflaviviruses, is not known. Of the mutations identified in JEV, the phenotype of E-E138K appeared to be not conserved in WNV, so assessing this mutation in USUV will provide further information as to whether the effect of this mutation is JEV-specific [19,48]. The E-E306K mutation neighbours a residue that also mutated to a positive charge in cell-culture-adapted TMUV (M304R), which resulted in marked attenuation of the virus [12].

Two USUV NS1 mutants were designed to attenuate the virus by removing the (predicted) N-linked glycosylation sites, which have a consensus amino acid sequence, NxS/T (where x is any amino acid except proline). Studies from other orthoflaviviruses show that removal of all the glycosylation sites by mutation is most strongly attenuating, and, while this did not result in viable or stable virus for DENV-2 this strategy is tolerated in other viruses, including another DENV serotype [23,25,26,49,50,51]. Our first USUV NS1 mutant virus (NS1-AAA; Figure 1) contains alanine substitutions at the N -residues of the three putative glycosylation motifs of the NS1 protein (N130A, N175A, N207A). A second NS1 mutant (NS1-QQAAA) was designed based on a WNV study that showed stronger attenuation and reduced reversion when additional mutations were present around the first glycosylation site (H131Q, T132A; Figure 1) [24].

A single mutation was incorporated into the USUV NS2A protein at a site that, in KUNV, disrupts formation of the NS1’ protein (NS2A-A30P; Figure 1) as well as abates the innate immune evasion role of the NS2A protein [28]. The attenuation was weaker when this mutation was introduced into WNV, and the attenuated phenotype appeared to be strain-dependent in JEV [32,33,34,35]. Assessing the strength of the effect of this mutation in USUV will help to answer how conserved the attenuation phenotype of this mutation is.

Three separate mutations in the USUV NS4B protein were also designed. NS4B-P41G was selected because the mutation of the corresponding residue in WNV significantly reduced lethality in animal models [37]. We also chose to introduce the NS4B-C102S mutation in USUV, as similar mutations in WNV and ZIKV were strongly attenuating [37,38]. An USUV mutant with the NS4B-D252G mutation was constructed because the corresponding NS4B-E249G substitution was identified in naturally attenuated variants of WNV. While this mutation was not attenuating in highly pathogenic WNV strains, there was evidence of reduced RNA synthesis in a WNV replicon. By including mutations in USUV with potentially mild as well as strongly attenuating phenotypes, we aimed to assess how well the strength of the attenuating phenotypes was retained [32,40].

For each of the above-mentioned mutations, we first determined whether the targeted amino acid was conserved in USUV and how broadly conserved the site was across a range of orthoflaviviruses (Appendix A). Where possible, we designed the amino acid substitutions in a way that minimised the likelihood of reversion, i.e., by selecting codons that require two nucleotide substitutions rather than a single substitution to revert.

### 3.2. Rescue and Stability of Mutant Viruses

Utilising a wild-type USUV-Af-3 full-length cDNA clone as a template, mutations were introduced using site-directed mutagenesis and TAR recombineering, as described in Section 2.2. The presence of the mutation(s) in rescued viruses was first confirmed by Sanger sequencing. Each mutant was then passaged four times on Vero CCL-81 cells after which RNA was isolated, reverse -transcribed, and analysed by NGS to assess the stability of the mutations and the possible emergence of second-site/compensating mutations elsewhere in the genome. Plaque phenotypes were also assessed (Appendix A).

The E-E138K mutation was still present in the passage 4 sequence, and no additional mutations were detected. The E-E306K mutation was present in approximately 80% of the reads, while the other variants in the population displayed reversion to Glu or had other negative or uncharged amino acids (Asp, Asn, Gln, His) at this position. Both of the NS1 glycosylation mutants were stable at all of the mutated sites, but the NS1-QQAAA mutant acquired two additional mutations, one at a low frequency at the amino acid directly after the third glycosylation site and one in the NS5 protein. The NS2A-A30P and NS4B-C105S mutations were stable, and no additional mutations were observed. A first attempt to launch the NS4B-P41G mutant virus failed to yield infectious virus, and a second attempt was successful, but the virus had low titres according to the plaque assay on BHK21-J cells and only caused variable and weak CPE on Vero CCL-81 cells. By passage 4, the engineered mutations were still present in ~80% of the NGS reads, but ~20% showed a substitution that resulted in a change to alanine at position 41. Additionally, the P4 virus stock contained a second mutation, T92P, in NS4B (73% of the reads). The NS4B-D252G mutant showed 75% reversion by passage 4. The stability results are summarised in Table 1.

### 3.3. Rationally Designed USUV Mutants Exhibit Varying Levels of Attenuation In Vitro

For each of the constructed USUV mutants, we assessed whether they displayed the expected attenuated phenotype by analysing their plaque morphology and replication kinetics on different cell lines and specific cell-based assays. Where relevant, we also assessed protein expression by Western blot or determined temperature sensitivity.

The USUV E-E138K mutant displayed a similar phenotype as the wild-type virus (Figure 2), in line with the observations for WNV [19]. USUV E-E306K had a reduced plaque size as well as replicated slower and to lower titres on Vero CCL-81 cells (Figure 2C) but reached higher titres than the wild-type virus on BHK21-J cells (Figure 2D). This difference in replication in these cell lines was in line with our expectations based on the results from previous studies on GAG-binding mutants [13,15,48]. Heparin is a proteoglycan that is structurally similar to heparin sulphate, a highly expressed GAG on mammalian cells. Viruses with enhanced GAG binding therefore have an increased sensitivity to inhibition by heparin. Unexpectedly, the wild-type USUV was already highly sensitive to inhibition by heparin, as was the E-138K mutant. The E-306K mutant on the other hand was less sensitive to low concentrations of heparin, showing a minimal reduction in the number of plaques formed when compared to the untreated control. Only concentrations of 20 μg/mL heparin or higher resulted in a reduced number of plaques for the E-306K mutant virus (Figure 2B).

The loss of NS1 glycosylation of the NS1-AAA and NS1-QQAAA mutants was confirmed with Western blotting (Figure 3A). Both mutants showed a strongly attenuated phenotype, with small, badly formed plaques that were difficult to distinguish and count (Figure 3B and Appendix A). The growth curves on Vero CCL-81 showed that the replication of both viruses was crippled, though the NS1-QQAAA mutant (containing additional amino acid changes in the first glycosylation motif) was more highly attenuated, barely causing CPE even at later timepoints, despite the increasing copies shown by RT-qPCR (Figure 3C,D).

The USUV NS2A-A30P mutant resulted in a loss of NS1’ expression, as shown by Western blotting (Figure 4A), as was observed for KUNV and JEV [29,34]. Even so, the mutant virus grew to a similar titre as the rUSUV-WT virus and showed no difference in plaque morphology (Figure 4B and Appendix A). On Vero CCL-81 cells, which are interferon-deficient, the mutant virus replicated similar to the wild-type virus (Figure 4C). However, this mutation was expected to influence viral innate immune evasion, and therefore we also constructed growth curves in A549 cells, an immunocompetent cell line [28,52]. In these cells, NS2a-A30P reached lower titres than the wild type (Figure 4D), although the effect was not as drastic as was observed for KUNV [28].

The USUV NS4B-P41G mutant displayed a small plaque phenotype (Figure 5A) and slower growth kinetics (Figure 5B). NS4B-C105S and NS4B-D252G showed a less dramatic, but statistically significant, decrease in plaque size, while they still replicated similar to the WT in Vero CCL-81 cells (Figure 5A,B). For the other orthoflaviviruses, the NS4B mutants with an attenuated phenotype in vivo were shown to be temperature-sensitive, and therefore we determined the fold reduction in the titre of the NS4B mutants at 41 °C compared to 37 °C. The P41G mutant was very temperature-sensitive and displayed an ~4 log reduction at 41 °C, while the wild-type virus only displayed a one log reduction at the higher temperature. Unexpectedly, the C105S mutant was not temperature-sensitive, while the D252G mutant showed a log reduction of approximately two in titre (Figure 5C).

### 3.4. Ifnar^−/−^ Mice Infected with Clone-Derived Wild-Type USUV Have a Marginally Longer Survival Time than Those Infected with the Natural Isolate 

All our mutant viruses were derived from the USUV-Af-3 full-length recombinant cDNA clone, and therefore we first compared the parental rUSUV-WT virus to the natural isolate USUV Af-3-NL in our animal model. Because immunocompetent mice do not support (robust) USUV infection after peripheral inoculation, we used Ifnar^−/−^ mice, which lack IFN α/β receptors. The mice were injected SC with clone-derived (rUSUV-WT) virus, natural isolate (USUV-Af-3) virus, or DMEM only. Mice were monitored for weight as well as clinical symptoms and bled on alternating days to assess viremia (Figure 6A). The mice infected with the clone-derived virus displayed a delay in weight loss and longer average survival time by approximately 1 day (Figure 6B,C). A single mouse from the rUSUV-WT-infected group showed a recovery of weight loss after day 7 (when all other mice had succumbed); however, this mouse became moribund and needed to be sacrificed by day 10. The increase in viremia was also delayed in the rUSUV-WT-infected mice, and the peak titres reached by the clone-derived virus were around 3-fold lower than those measured for the corresponding natural isolate (Figure 6D). Reduced virus titres were also found in the tissues harvested at the humane endpoint (HEP), showing a 3- to 10-fold reduction in viral copy numbers across the heart, liver, spleen kidney, and brain (Figure 6E).

We consistently observed this slight delay in lethality across numerous studies comparing rUSUV-WT-infected mice compared to natural isolate infected mice (Appendix A). Based on these results, we used the clone-derived rUSUV-WT virus as the control for assessing the phenotypes of our recombinant mutant viruses.

### 3.5. Five of Six Mutant Viruses Cause Decreased Mortality and Increased Average Survival Time in Ifnar^−/−^ Mice

The mutations that caused attenuated phenotypes in vitro (E-E306K, NS1-AAA, NS1-QQAAA, NS2A-A30P, NS4B-P41G, NS4B-D252G) were assessed in the Ifnar^−/−^ mouse model (Figure 7A). With the exception of the NS2A-A30P mutant, the attenuating mutations all caused statistically significant increases in survival percentage and average survival time and delayed the onset of clinical symptoms in comparison to the animals inoculated with the wild-type virus (Figure 7B,C, Table 2).

The highest level of attenuation was seen with the two NS1 glycosylation mutants. All mice infected with NS1-AAA survived and showed minimal clinical symptoms—a limp in the injected limb in six out of eight mice and around 5% weight loss (Figure 7B,C and Table 2). The weight loss was delayed compared to the rUSUV-WT controls by several days, and weight began to increase again by day 7. The NS1-QQAAA-mutant-infected group showed no weight loss or other clinical symptoms, with the exception of a single animal. This mouse started to noticeably lose weight around day 7, developed a limp by day 8, and progressively declined in health until HEP was reached on day 12 (Table 2, Appendix A).

Only a single animal succumbed in the NS4B-P41G-infected group as well, although all the mice showed weight loss from day 5, which peaked around day 7. Correlating with the reduced weight, the mice also had reduced activity levels and a hunched posture (Figure 7B,C and Table 2). These mice then all recovered their condition, although some more rapidly than others (Figure 7B and Appendix A). The one mouse that was sacrificed had an ocular discharge in both eyes from day 7 and developed paralysis of the hindlimbs on day 9. For the NS4B-D252G-infected mice, a delay in the onset of weight loss was also observed, even more so than in the other mutant-infected groups. Half the mice did not develop clinical symptoms beyond a mild weight loss, from which they quickly recovered; however, four of these animals became progressively more moribund and developed paraplegia or tetraplegia on day 10 and 11 and were put down.

The GAG-binding mutant E-306K also resulted in lethality of 50% of the animals (Figure 7B). All mice in the group developed clinical symptoms, starting to lose weight around day 4 to 5 and becoming lethargic several days later (Figure 7C, Table 2). One mouse deteriorated in condition rapidly, reaching HEP by day 7, and three more mice were sacrificed on days 9 and 10 due to paralysis. The four remaining animals however had almost fully recovered by this point and continued to gain weight until the end of the experiment (Appendix A).

Lastly, the NS2A-A30P mutation had no effect on the survival or clinical symptoms of the mice compared to the rUSUV-WT control (Figure 7B,C). A single mouse survived until day 9 and was recovering weight, when it developed paralysis and was sacrificed (Appendix A, Table 2). However, this increase in survival time was not statistically significant and was also observed previously with rUSUV-WT-infected animals (Appendix A).

### 3.6. Virus Is Detectable in Tissue and Sera from Surviving Mice

Sera were collected from the mice on alternating days to compare the viremia in the mice (Figure 8A). In keeping with the trends observed above, the onset of measurable viremia was delayed for the mutant groups except for NS2A-A30P-infected mice. While the rUSUV-WT titre was already measurable on day 3 and remained high until the mice succumbed on day 6, in most of the mutant-infected mice, the viremia was only detectable by day 4 or even day 5. Furthermore, the peak titre for these groups occurred later and was significantly lower than for rUSUV-WT -infected animals (*p* = 0.05 or lower). Except for the single animal that died, no mice in the NS1-QQAAA group developed measurable viraemia. While the viremia decreased in most groups by day 7 or 8, the NS4B-P41G and NS4B-D252G groups both had measurable viremia past day 8 or 9 in around half the animals. In the NS4B-D252G mice, this was predictive of the animals that eventually reached HEP.

The liver, spleen, kidney, and brain were harvested at HEP, viral load was determined by RT-qPCR, and TCID50 assays were performed on all RT-qPCR-positive samples. Positive titres were detected in the brain of all the mice that succumbed to disease, indicating that these viruses were neuroinvasive (Figure 8B). The other tissues were mostly negative by this point, fitting with the occurrence of HEP some time after the peak (serum) viremia, as infection of the brain was a later-stage phenomenon.

### 3.7. Some Mutations Are Less Stable and Show Partial Reversion in Mice, Which May Correlate to Observed Lethality Rates 

For some of the mutants, part of the group of infected mice survived, while others succumbed to the infection. We hypothesised that in those mice that died, the virus may have reverted to the wild type, causing neuroinvasion and consequently death. We therefore performed NGS on the cDNA from RNA isolated from the brain tissues (harvested at HEP) of the mice that succumbed (Table 3). USUV E-E306K was lethal in four out of the eight mice. Two of the succumbed mice indeed showed reversion from the Lys residue back to wild-type Glu, while the other two mutated to an Asn, an uncharged amino acid. A single NS1-QQAAA-dosed mouse succumbed to infection 12 days after infection, and the viral RNA from this mouse still had all of the introduced mutations. Three mice were selected for sequencing from the NS2A-A30P group, and no reversion was observed. The single NS4B-P41G-infected mouse that succumbed by day 9 also did not show any reversion.

We were unable to obtain sequencing reads from three of the NS4B-D252G mice that succumbed; however, the viral cDNA from the brain of one mouse (mouse 41) was sequenced, and this showed reversion to the wild type in 45.5% of the reads. To obtain results for the other NS4B-D252G mice, we generated a smaller cDNA of the mutated region, which was analysed by Sanger sequencing. Two mice (mice 40, 41) showed reversion in approximately 50% of the sequences, one mouse (mouse 42) showed about 25% reversion, and one mouse (mouse 44) retained the original mutation.

To further assess whether reversion was occurring, we analysed the plaque phenotypes of the viruses from homogenised brain tissue. The viruses isolated from the samples that had reverted to the wild type should have had plaque sizes more similar to the rUSUV-WT control, while the virus samples without reversion should have retained the phenotype of the mutant stock virus. The plaques from the rUSUV-WT-infected animals were around 80% of the size of the stock virus (Figure 9A). The two E-E306K animals that mutated to an Asn had a smaller plaque phenotype than the stock virus (60% relative plaque size compared to 80% relative plaque size), while the Lys to Glu revertant samples showed a similar plaque phenotype to the rUSUV-Af3-infected mice (~80% relative plaque size) (Figure 9B). The NS1-QQAAA brain plaques from the single mouse that succumbed in this group were larger than the mutant rNS1-QQAAA stock but still much smaller than the WT virus stock (Figure 9C). The NS2A-A30P virus stock plaque size was similar to the wild-type virus, and, consistent with this, the brain tissue plaques showed no change in phenotype (Figure 9D). The P41G-infected animals showed the same plaque size as the mutant stock, which was ~60% of the wild-type virus (Figure 9E). The plaques from the four D252G-infected animals appeared to maintain the mutant virus plaque phenotype (Figure 9F).

We were unable to assess whether there was also reversion or additional mutations in the mice that survived the experiment due to the (too) low virus titres in these animals, samples of which were harvested only by the end of the experiment. The serum samples taken at peak viremia could show whether mutations/reversion already occurred early on, but, due to the low volume, sequencing was not possible.

### 3.8. Infection with Attenuated Mutant Viruses Results in High Neutralising Antibody Titres

To assess the immune response to the mutant viruses, we performed virus neutralisation assays on the sera harvested from all mice at the time they were sacrificed (HEP or at the end of the experiment). Surprisingly, we detected high titres of neutralising antibodies even at early timepoints, including for the rUSUV-WT-infected animals, which succumbed by day 6. For the viruses with an attenuated phenotype (E-E306K, NS4B-P41G, NS4B-D252G), these titres were consistently higher for mice that succumbed to mutant virus infection than for mice that survived and were harvested at the end of the experiment (Figure 10).

## 4. Discussion

Based on knowledge from other orthoflaviviruses, we rationally designed a variety of mutations in USUV, aiming to attenuate the virus and determine if the mechanisms of attenuation are broadly conserved. The mutants were assessed in vitro and then put into a mouse model to determine whether the (attenuated) phenotype was conserved. While some of the designed mutations were strongly attenuating, as expected, other mutations resulted in less-expected/predictable phenotypes, probably partly due to the instability of the inserted mutations in our experiments (Table 4).

The USUV E-E138K mutation did not confer the anticipated attenuating phenotype in vitro and was therefore not tested in the animal model. This was not completely unexpected, as this mutant phenotype was also not conserved in WNV [19,48]. The USUV E-E306K phenotype was attenuating, though less strong than has been observed for JEV or the corresponding mutation in TUMV [12,19]. Aligning with previous studies of GAG-binding phenotypes, we observed a small plaque phenotype, increased replication titres in BHK21-J cells, and enhanced survival in vivo. However, in contrast to the previous literature, the USUV E-E306K mutant appeared to be less sensitive to inhibition by heparin than the wild-type virus, as the latter was strongly inhibited even at low concentrations of heparin [11,13,14]. In earlier studies, GAG-binding mutations were identified by reverse genetics from viruses displaying an attenuated phenotype after serial passaging in specific cells lines (such as BHK-21, SW13). The identified mutations are located in varying regions within the envelope protein for different viruses, and, our results, as well as earlier WNV studies, show that these identified mutations may be virus-specific [13,14,15]. There may be GAG-binding-site mutations that are more stable or more reliably confer an attenuating phenotype in a broad range of viruses, but more research is required to assess this. Instead of the site-directed approach, serially passaging USUV or other emerging viruses on a relevant cell line to obtain mutations that affect GAG binding and attenuate the virus might be a more efficient tactic. Therefore, while this mechanism is highly conserved, the rational design of specific mutations to modulate GAG binding and confer the attenuating phenotype is not easily applicable to vaccine design for emerging viruses.

The removal of the NS1 glycosylation sites has a strongly attenuating effect in USUV. Similar to observations in other orthoflaviviruses, the glycosylation mutants displayed crippled replication in vitro and dramatically reduced pathology in a mouse model [10,23,24,25,26,27]. As seen in WNV, the NS1-QQAAA mutant, containing additional mutations in the first glycosylation motif, was more strongly attenuated than the NS1-AAA mutant [24]. No viremia or disease was detected for seven out of eight NS1-QQAAA-inoculated mice, but the neutralising antibody titres in the sera confirmed that an infection occurred. In the one mouse for which NS1-QQAAA infection was lethal, USUV was detected in the sera and brain tissue by RT-qPCR, although no infectious virus was measured in the brain with the TCID50 assay. The clinical symptoms of this mouse were similar to those of rUSUV-WT-infected animals but with a delayed onset; however, sequencing confirmed there was no reversion to the wild type.

The extent of attenuation caused by mutating NS1 glycosylation sites may be virus-specific (although differences in methods and models used could also explain the variations that were observed). In contrast to the results in WNV and USUV, incorporating the additional mutations in the two glycosylation motifs of DENV-4 was actually less attenuating than including just the Gly to Ala mutations at both sites [26]. In DENV-2, the substitution of Gly to Ala at both sites resulted in either an extremely unstable virus or a non-viable virus [49,53]. In YFV and TMUV, the removal of all NS1 glycosylation sites resulted in strong attenuation in vivo [23,25]. The YFV and DENV studies both assessed neurovirulence (via intracranial inoculation), so whether there is a reduction in neuroinvasion (using peripheral inoculation) is not known [23,26,49]. Despite the overall conservation of attenuation of NS1 glycosylation mutants, these results show that which amino acid changes are most effective in a vaccine candidate should be assessed carefully per virus.

The USUV NS2A-A30P mutant disrupted NS1’ production but was only mildly attenuated in our in vitro studies and showed no phenotype in the Ifnar^−/−^ mouse model. This aligns with the results for WNV and JEV and indicates that the attenuation effect of NS2A-A30P in KUNV is not very well conserved [28,32,33,34]. The NS2a-A30P mutation has been shown to act via multiple roles: the disruption of the innate immune evasion function of the NS2A protein and the loss of the NS1’ protein [28,52]. Our use of an immunocompromised animal model means that any mechanism of attenuation relying on disruption of IFNα/β antagonism could not be observed. In KUNV, however, attenuation occurred even in the Ifnar^−/−^ mice (although less so than in the immunocompetent mice), implying a mechanism of attenuation apart from the disruption of the innate immune antagonism that is not conserved in USUV [29]. In immunocompetent A549 cells, the USUV NS2A-A30P mutant was only mildly attenuating, while the KUNV NS2A-A30P mutation severely crippled replication. Therefore, we predict that even with an immune competent mouse model for USUV, this mutation might only cause a weak attenuation.

The USUV NS4B-P41G mutation was strongly attenuating, similar to observations in WNV [36]. The mechanism for this attenuation in WNV was linked to reduced antagonism of the innate immune response based on the increased stimulation of type 1 interferons and IL-1β by the NS4B-P38G mutant compared to the wild-type virus in a mouse model [36]. As the Ifnar^−/−^ mouse model lacks receptors to α/β IFN (type 1 IFNs), our results imply there may be additional or alternative mechanism for the attenuation we observed or that the attenuation could be stronger in wild-type hosts.

The USUV NS4B-C105S mutation had only a mild effect in our in vitro models, in contrast to observations in WNV and ZIKV [37,38], and this mutant was not carried forward into our in vivo studies. The mechanism of attenuation for this mutation is not well understood. While it was initially designed based on the predicted importance of the cysteine residue, the substitution of the Cys for an Ala was less strongly attenuating, implying the attenuation was not solely due to the loss of the stabilising cysteine bond [38,39]. Studies on ZIKV show that the NS4B-C100S mutation induces higher IFN-α, IFN-β, and IL6 mRNA levels, indicating a disruption of innate immune antagonism [38]. However, more studies are needed to understand why mutating this amino acid may play less of a role in USUV and whether the attenuating effect might be better conserved in more pathogenic orthoflaviviruses.

The USUV NS4B-D252G mutation was moderately attenuating in vitro and resulted in higher survival than in the wild-type virus infections, reflected in half the animals surviving. The substitution was unstable, showing partial reversion to the wild-type virus in the passage four virus isolate as well as in three out of four mice that succumbed. NS4b-D252 may therefore be quite an important virulence determinant in USUV. The importance of this site in USUV is somewhat surprising given that the NS4b-D252G phenotype, identified in a WNV isolate, was not conserved in a second WNV strain [32,40]. The NS4b-C105S mutation, on the other hand, despite being attenuating in WNV and the more distantly related ZIKV, was not attenuating in USUV. These cases further highlight how utilising virulence-determinant information from related orthoflaviviruses may not be an efficient way to design attenuating mutations for emerging viruses.

There are several considerations that should be kept in mind when assessing these data. First, the defects in the innate immune response of the Ifnar^−/−^ mouse model result in an altered pathology of the virus infection relative to the situation in wild-type hosts. As discussed above for specific mutants, this provides a significant limitation in the ability to study attenuating mutations in cases where the mechanism involves the IFNα/β pathways. Immunocompetent mice however are not reliably susceptible to USUV infection and can therefore not be used to assess infection [54]. While progress is being made in alternative disease models for orthoflavivirus research, mouse models are unfortunately still the gold standard and the most reliable way to assess virus pathogenicity [55].

Second, we sought to assess the impact of single mutations (apart from the combination of NS1 glycosylation mutations) on USUV pathogenicity. Therefore, few mutations are required in order for the mutant to revert to the wild type, and indeed we observed reversion in a number of the mutants. To create a safe live attenuated vaccine candidate, it will be important to combine a number of rationally designed mutations and to carefully assess the reversion potential of any candidate vaccines.

Third, this study was performed using an isolate from the Africa-3 USUV lineage. It is clear from the literature however that the phenotypes of many of these mutations can differ even within a single virus species [26,32,33,56]. Whether these results would translate to other USUV lineages, especially a more virulent lineage like Eu-2, requires further research [57,58]. Additionally, there is evidence that the site of inoculation appears to play a role in the disease outcome and pathogenicity that may be specific to this Africa-3 isolate, although this is not yet well understood [44,59]. Therefore, there may be methodological or isolate-specific factors in our study that influenced the attenuating phenotype of these mutations that could be observed, for example, based on the immune response of the specific cells in the different inoculation sites or on the relative neuroinvasive or neurovirulent capacity of the different isolates.

## 5. Conclusions

Of the eight mutations we selected from the literature that had been assessed in other orthoflaviviruses that we here transferred into USUV, only five were attenuating in an in vivo model. Furthermore, neither the mechanism driving the attenuation nor the conservation of the phenotype in other viruses was reliably predictive of the outcome of USUV infection. While these mutations give us insight into some of the virulence determinants of USUV, understanding which mutations may be most promising to incorporate into future vaccine designs requires a deeper understanding of how different mutations impact different viruses. In USUV, NS1 glycosylation mutants may be overly attenuated, and GAG-binding enhancement mutations may have potential, but more work on specific sites needs to be conducted. Mutations in the N-terminal innate immune motif in NS4B are potentially useful, showing conservation in USUV as well as ZIKV and WNV; however, single mutations are at risk for reversion and would need to be combined with other safety features in order to make a safe vaccine candidate. Overall, we conclude that while safe-by-design vaccines are promising, their application may be better suited for well-characterised viruses. Deeper research is required to apply this approach to future emerging orthoflaviviruses.

## Figures and Tables

**Figure 1 vaccines-13-00495-f001:**
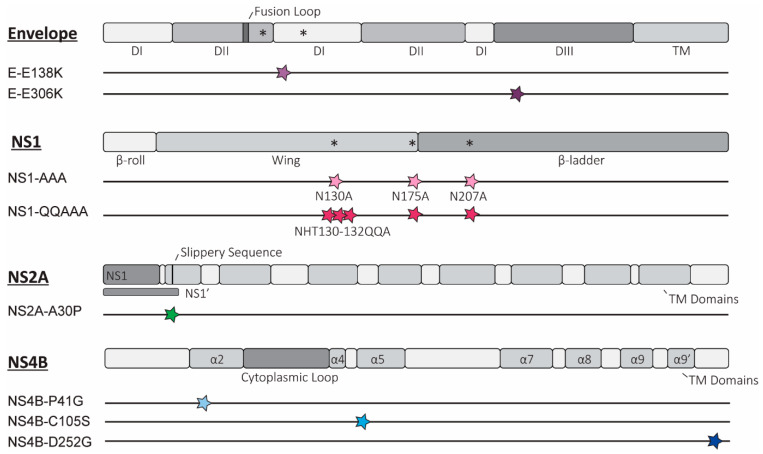
Overview of rationally designed mutations introduced into Usutu virus (USUV). Schematic representation depicting the location of the potentially attenuating mutations in the relevant proteins for each of the 8 mutant viruses, as elaborated in the text. The key domains and structural motifs are depicted for the envelope (E) proteins and non-structural proteins 1 (NS1), 2a (NS2A), and 4b (NS4B) (individual proteins are not depicted to scale based on relative sizes). For each of the constructed mutant viruses, the site of the mutation(s) is shown as a coloured star, correlating to colours used to represent each mutant in later figures. * = N-linked glycosylation site; TM = transmembrane region; DI/II/III = domain I/II/III; α = alpha helix; other structural or sequence elements are indicated with text.

**Figure 2 vaccines-13-00495-f002:**
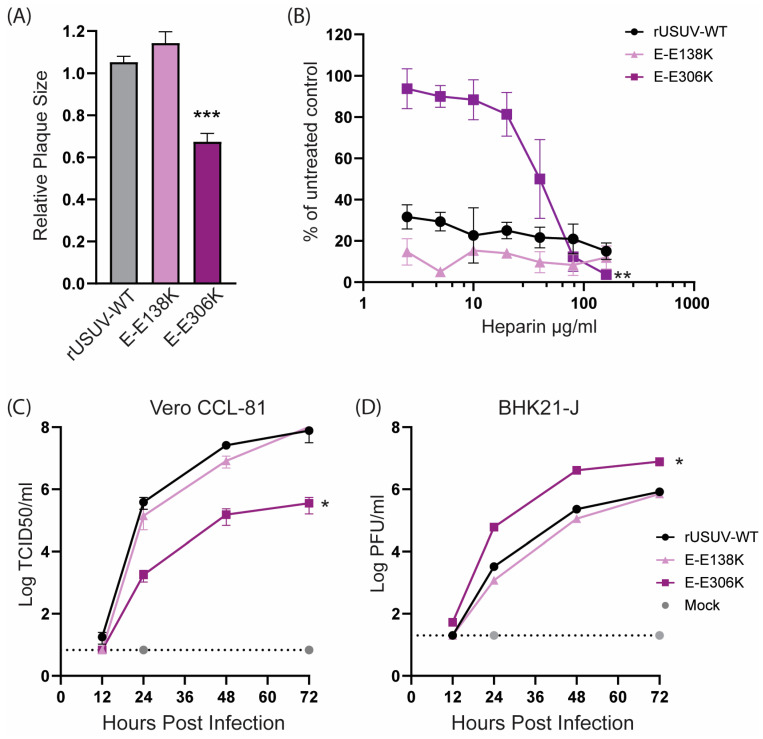
Cell culture phenotype of the engineered USUV envelope mutants rationally designed to enhance glycosaminoglycan (GAG) binding. (**A**) Plaque diameters of mutant viruses normalised to the average diameter of the rUSUV-WT plaques on BHK21-J cells. Statistical analysis was performed using unpaired *t*-test. (**B**) Heparin inhibition in a dose–response assay on BHK21-J cells. Sensitivity of the inhibition is shown as percentage of plaques formed in treated cells relative to untreated control. Replication kinetics on (**C**) Vero CCL-81 and (**D**) BHK21-J cells infected with rUSUV-WT or E-E138K and E-E306K mutant viruses. Infectious virus titres in cell culture supernatants harvested at specified timepoints were determined by TCID50 or plaque assay. Statistical analysis was performed using one-way ANOVA. * *p* < 0.05, ** *p* < 0.01, *** *p* < 0.001.

**Figure 3 vaccines-13-00495-f003:**
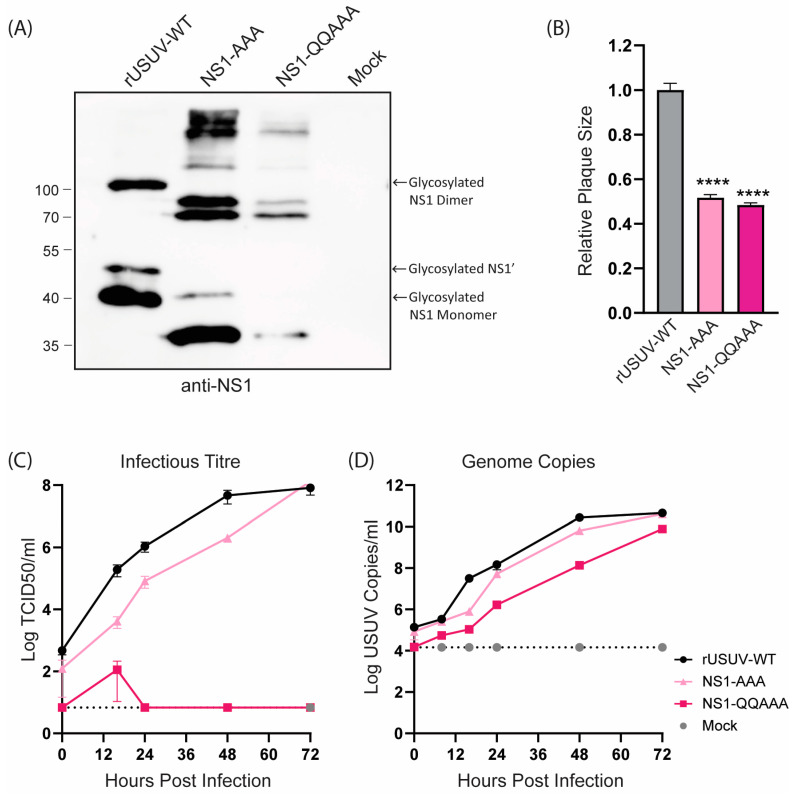
Phenotype of engineered USUV NS1-glycosylation mutants in cell culture. (**A**) Western blot of protein lysates harvested at 48 hpi from Vero CCL-81 cells infected with rUSUV-WT or mutant viruses treated with anti-NS1 antibodies. (**B**) Plaque diameters of mutant viruses normalised to average diameter of rUSUV-WT plaques on BHK21-J cells. Statistical analysis was performed using unpaired *t*-test. Replication kinetics on (**C**) Vero CCL-81 and (**D**) A549 cells infected with rUSUV-WT or NS1-AAA and NS1-QQAAA mutant viruses. Titres were determined by TCID50 or plaque assay on cell culture supernatants harvested at specified timepoints. **** *p* < 0.0001.

**Figure 4 vaccines-13-00495-f004:**
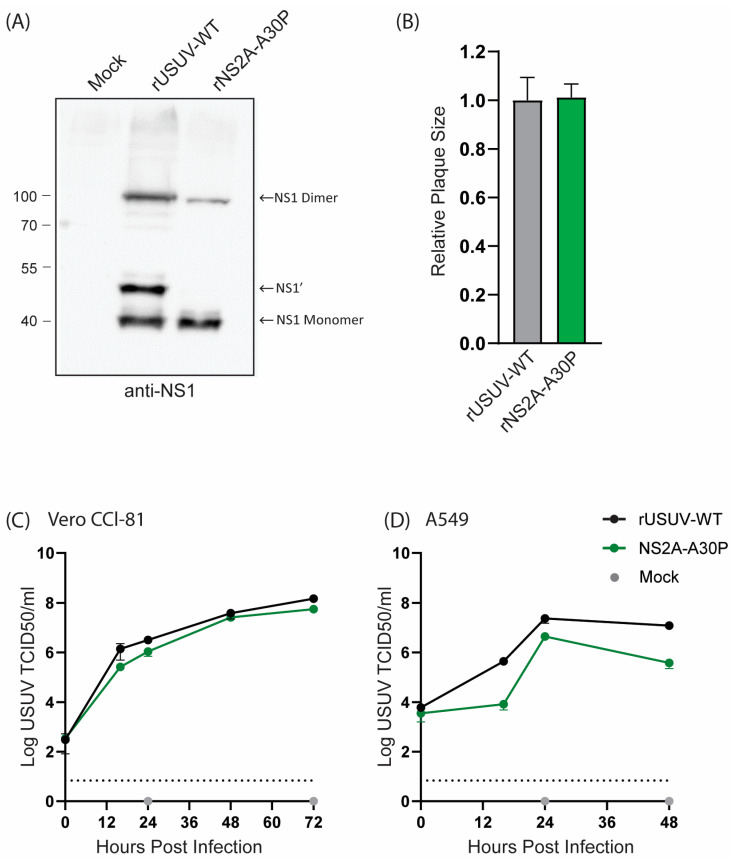
Cell culture phenotype of USUV NS2A-A30P mutant. (**A**) Detection of NS1’ and NS1 protein from Western blot of 48 h lysates from Vero CCL-81 cells infected with rUSUV-WT or mutant virus. (**B**) Plaque diameters of mutant viruses normalised to average diameter of rUSUV-WT plaques on BHK21-J cells. Statistical analysis was performed using unpaired *t*-test. Replication kinetics on (**C**) Vero CCL-81 and (**D**) A549 cells infected with rUSUV-WT or NS2A-A30P mutant virus. Titres were determined by TCID50 assay from cell culture supernatants harvested at specified timepoints.

**Figure 5 vaccines-13-00495-f005:**
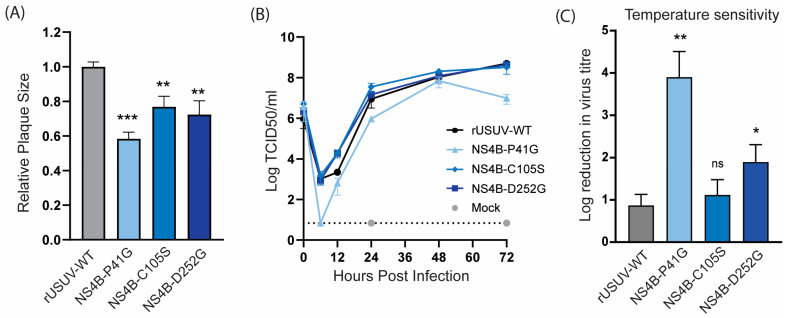
Cell culture phenotype and temperature sensitivity of USUV NS4B mutant viruses. (**A**) Plaque diameters of mutant viruses normalised to the average diameter of rUSUV-WT plaques on BHK21-J cells. Statistical analysis was performed using unpaired *t*-test. (**B**) Replication kinetics on Vero CCL-81 cells infected with rUSUV-WT or NS4B-P41G, NS4B-C105S, and NS4B-D252G mutant virus. Titres were determined by TCID50 assay from cell culture supernatants harvested at specified timepoints. (**C**) TCID50 assays were performed with rUSUV-WT or NS4B-P41G, NS4B-C105S, and NS4B-D252G mutant virus stocks on Vero CCL-81 at either 37 °C or 41 °C. Temperature sensitivity is expressed as reduction in virus titre at high temperature (41 °C) compared to standard growth temperature (37 °C) for each virus. Statistical analysis was performed using unpaired *t*-test. ns = not significant, * *p* < 0.05, ** *p* < 0.01, *** *p* < 0.001.

**Figure 6 vaccines-13-00495-f006:**
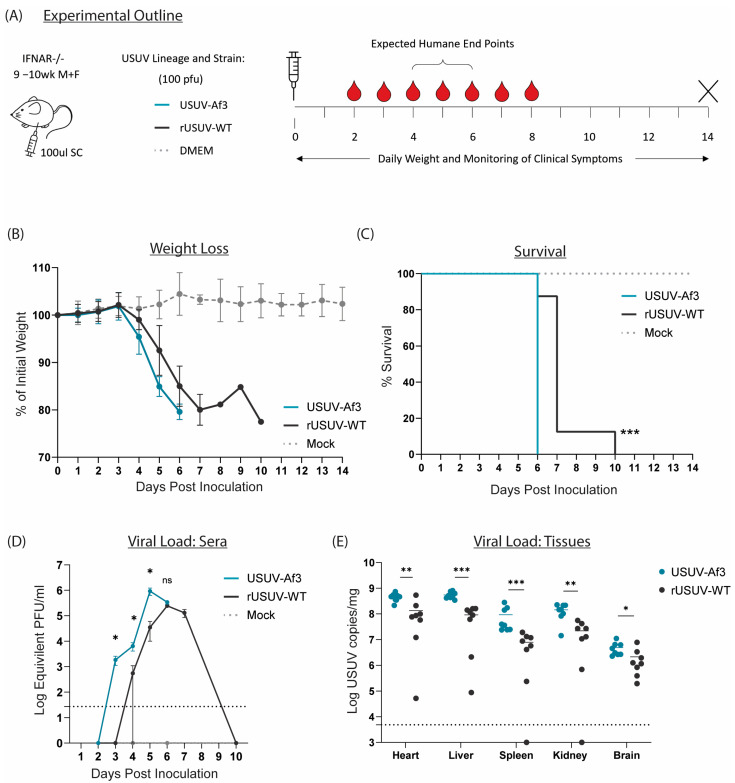
Comparison of recombinant-clone-derived USUV to its corresponding natural isolate in a mouse model. (**A**) Ifnar^−/−^ mice (9-10wk) were inoculated SC with 1 × 100 pfu/mouse of clone-derived (rUSUV-WT) virus or corresponding natural isolate (USUV-Af3). They were weighed daily, and half of mice per group were tail-bled on alternate days. Animals were euthanised when they reached humane endpoint, final bleeds were taken by heart puncture, and relevant tissues were harvested. (**B**) Daily weight loss measured as a percentage of initial weight for each experimental group shown as mean ± SD. (**C**) Survival rates for each experimental group. Statistical analysis was performed using the log-rank (Mantel–Cox) test. (**D**) Mean (±SD) viral load measured by RT-qPCR of tail bleeds and final heart bleed sera using serial dilutions of reference standard to determine pfu equivalents. Statistical analysis was performed using unpaired *t*-test for each timepoint. (**E**) USUV RNA copies/mg of homogenised heart, liver, spleen, kidney and brain tissues harvested at humane endpoint or end of experiment (day 14) measured by RT-qPCR and absolute quantification using reference standard. Statistical analysis was performed using unpaired *t*-test corrected for multiple analyses. Limit of detection presented as dotted grey line. ns = not significant, * *p* < 0.05, ** *p* < 0.01, *** *p* < 0.001.

**Figure 7 vaccines-13-00495-f007:**
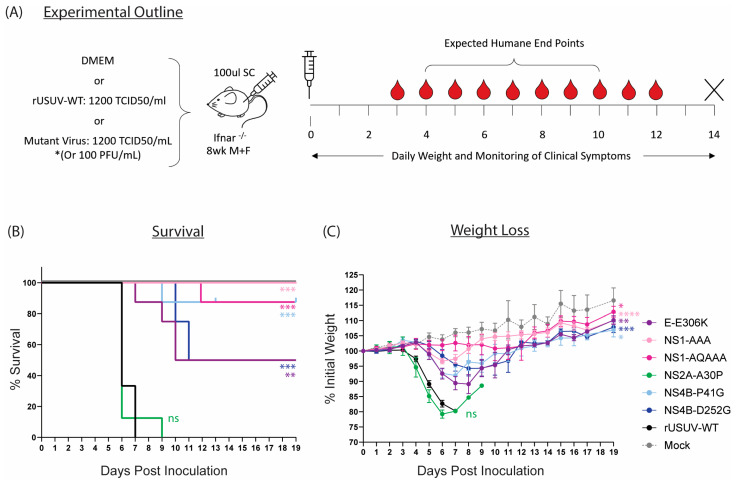
Characterisation of rationally designed USUV mutants in Ifnar^−/−^ mouse model. (**A**) Ifnar^−/−^ mice (6–8 wk) were inoculated SC with DMEM alone, 1.4 × 10^2^ TCID50/mouse of clone-derived rUSUV-WT virus, or 1.4 × 10^2^ TCID50/mouse of each mutant virus (or 100 pfu/mouse for the NS4B-P41G mutant). Animals were weighed daily, and half of mice per group were tail-bled on alternate days. Animals were euthanised when they reached humane endpoint, final bleeds were taken by heart puncture, and relevant tissues were harvested. (**B**) Kaplan–Meier curve showing percentage survival for each of the experimental groups. Statistical analysis was performed using log-rank (Mantel–Cox) test. (**C**) Average daily weight loss, measured as percentage of initial weight, for each experimental group. Statistical analysis was performed using mixed-effects models with Geisser–Greenhouse correction. ns = not significant, * *p* < 0.05, ** *p* < 0.01, *** *p* < 0.001, **** *p* < 0.0001.

**Figure 8 vaccines-13-00495-f008:**
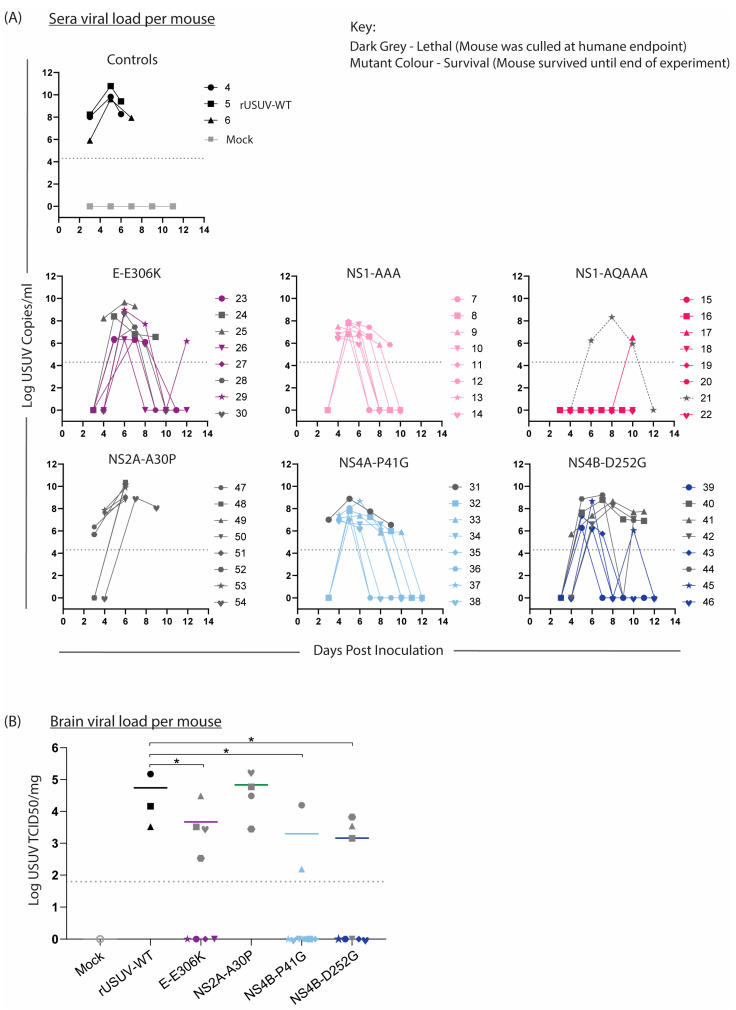
Viral loads in sera and brain tissue of mice infected with WT and mutant USUV. Bleeds and relevant tissues were harvested from Ifnar^−/−^ mice infected with rUSUV-WT or mutant viruses, as described in Figure 7. (**A**) Viral titre of tail bleeds and final heart bleed sera from control (rUSUV-WT and mock) and mutant-virus-infected mice, measured by RT-qPCR and absolute quantification using a reference standard. (**B**) USUV TCID50/mg of homogenised brain tissues harvested at humane endpoint (HEP) or end of experiment. Statistical analysis was performed using Mann–Whitney test. Limit of detection represented as dotted grey line. * *p* < 0.05.

**Figure 9 vaccines-13-00495-f009:**
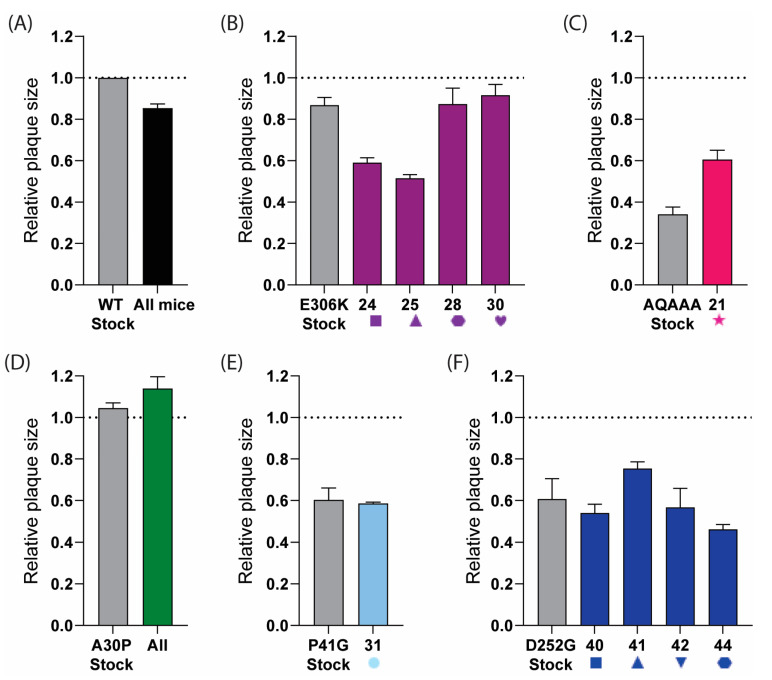
Plaque phenotype of USUV mutants as indicator for reversion in samples obtained from brains of mice that succumbed to infection. Plaque assays were performed on homogenised brain tissue from (**A**) rUSUV-Af3-inoculated control mice (graph shows combined data of three individual animals, or from individual mice that succumbed to infection with different mutant viruses. Individual mouse number is shown on x-axis for each group: (**B**) E-E306K, (**C**) NS1-QQAAA, (**D**) NS2A-A30P, (**E**) NS4B-P41G, and (**F**) NS4B-D252G D252G (shape and number for individual mice correlate to Figure 8). Stock virus for wild-type rUSUV-Af3 and for each mutant was taken as control per group (in grey). Data are shown as average ± SEM of plaque size of three wells per sample relative to plaques from stock aliquot of wild-type rUSUV-Af3 virus (dotted line) for 3 wells per group.

**Figure 10 vaccines-13-00495-f010:**
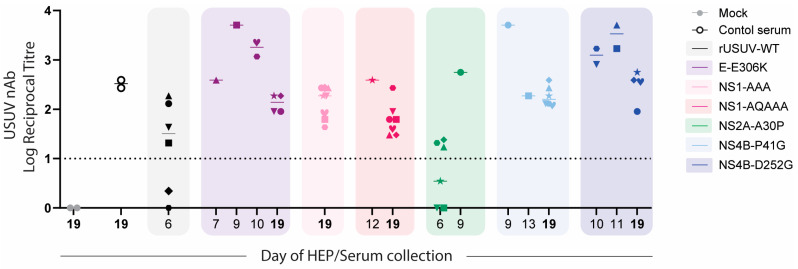
USUV neutralisation by sera obtained from mice infected with WT or mutant USUVs. Virus neutralisation assays were performed on sera collected from mice at HEP or from mice at end of experiment (day 19, in bold). GMTs (geometric mean titres) of neutralising responses against USUV are shown for mock and positive control sample (pooled serum sample collected from USUV-Af3 infected Ifnar^−/−^ mice on day 14, n = 2), rUSUV-Af3 (2 replicate experiments), and each of mutant viruses. Neutralisation titres at timepoints indicated on x-axis are shown for each mutant (coloured boxes). Symbol shape represents each individual mouse (1 through 8 per group) as shown in Figure 8.

**Table 1 vaccines-13-00495-t001:** Stability of inserted mutations in cell culture.

Mutant Virus (Amino Acid Substitution(s))		Present in P1	Present in P4	Additional Mutations
**E-E138K**		Y	Y (98.6%)	
**E-E306K**		Y	Y (80.3%)	
**NS1-AAA**	**N130A** **N175A** **N207A**	Y Y Y	Y (99.4%) Y (99.7%) Y (99.6%)	
**NS1-QQAAA**	**N130A** **H131Q** **T132A** **N175A** **N207A**	Y Y Y Y Y	Y (99.7%) Y (99.7%) Y (98.4%) Y (99.4%) Y (99.4%)	NS1—T208P (20%) NS5—A39S (99.8%)
**NS2A-A30P**		Y	Y	
**NS4B-P41G**		Y	Y (80.2%) (P41A 19.7%)	NS4B—T92P (73.2%)
**NS4B-C105S**		Y	Y	
**NS4B-D252G**		Y	25% only.	D252G only 25%

P1 = passage 1 virus stock, P4 = passage 4 virus stock. Y = yes, present. Numbers in brackets indicate the percentage of NGS reads that contained the sequence encoding the indicated amino acid. Silent mutations or mutations present at <10% were excluded—see Appendix A for full sequencing results.

**Table 2 vaccines-13-00495-t002:** Fraction of group showing clinical symptoms and median day of symptom onset for mutant-infected Ifnar^−/−^ mice.

	Clinical Symptom:									
Mutant	Reduced Activity ^1^		Hunched Posture		Limping ^2^		Paralysis ^3^		Ocular Discharge ^4^	
**rUSUV-WT**	3/3	Day 6	3/3	Day 6	3/3	Day 5	0/3	NA	3/3	Day 6
**E-E306K**	8/8	Day 9	4/8	Day 8	8/8	Day 5	3/8	Day 10	2/8	Day 8
**NS1-AAA**	0/8	NA	0/8	NA	6/8	Day 6	0/8	NA	0/8	NA
**NS1-QQAAA**	1/8	Day 11	1/8	Day 11	1/8	Day 8	0/8	NA	1/8	Day 10
**NS2A-A30P**	8/8	Day 6	8/8	Day 5	8/8	Day 5	1/8	Day 9	8/8	Day 6
**NS4B-P41G**	5/8	Day 7	5/8	Day 7	8/8	Day 6	1/8	Day 9	1/8	Day 7
**NS4B-D252G**	4/8	Day 8	4/8	Day 8	4/8	Day 6	4/8	Day 11	0/8	NA

^1^ Score when animals were no longer running around cage unprompted. ^2^ Limp developed in the inoculated hind limb. ^3^ Full paraplegia or tetraplegia or sufficient loss of limb use so that the animal was immobile. ^4^ White discharge around eye or partially/fully closed eye.

**Table 3 vaccines-13-00495-t003:** Sequencing of mutant USUV in brains of mice that succumbed to the infection.

	NGS Results	Sanger Results
Mouse Group(and Individual *)	Nucleotide Change	Percentage of Reads	AA Substitution	Nucleotide Change	Approximate Percentage Reversion	AA Substitution
**E_E306K ( 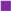 24)**	G1893C	100%	K306N	N/A
**E_E306K ( 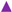 25)**	G1893C	100%	K306N	N/A
**E_E306K ( 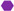 28)**	A1891G	100%	K306E	N/A
**E_E306K ( 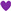 30)**	A1891G	100%	K306E	N/A
**NS1_QQAAA ( 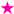 21)**	No reversion	N/A
**NS2A_A30P ( 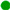 47)**	No reversion	N/A
**NS2A_A30P ( 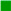 48)**	No reversion	N/A
**NS2A_A30P ( 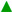 49)**	No reversion	N/A
**NS4B_P41G ( 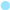 31)**	No reversion	N/A
**NS4B_D252G ( 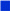 40)**	No read coverage	G7664A	~50%	G252D
**NS4B_D252G ( 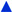 41)**	G7664A	45.5%	G252D	G7664A	~50%	G252D
**NS4B_D252G ( 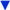 42)**	No read coverage	G7664A	~25%	G252D
**NS4B_D252G ( 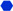 44)**	No read coverage	none	0%	N/A

* Shape and number correlate to Figure 8.

**Table 4 vaccines-13-00495-t004:** Summary of the phenotypes of the rationally designed mutations in USUV.

Mutation in USUV	In Vitro Phenotype	In Vivo Phenotype	Suitability for Inclusion in Future Vaccine Candidates
**E-E138K**	No attenuation	n/a	No—attenuation not conserved
**E-E306K**	Moderate attenuation	Partial attenuation	Low—reversion risk
**NS1-AAA**	Strong attenuation	Strong attenuation	Possible
**NS1-QQAAA**	Strong attenuation	Strong attenuation	Possible
**NS2A-A30P**	Mild attenuation	No attenuation	Low—attenuation not conserved
**NS4B-P41G**	Moderate attenuation	Strong attenuation	Possible
**NS4B-C105S**	Mild attenuation	n/a	Low—attenuation not conserved
**NS4B-D252G**	Moderate attenuation	Partial attenuation	Low—reversion risk

n/a—not applicable (in vivo study not performed).

## Data Availability

No datasets were generated or analysed during the current study.

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
