# Peer review of "Attenuating Mutations in Usutu Virus: Towards Understanding Orthoflavivirus Virulence Determinants and Live Attenuated Vaccine Design"

_vaccines, 2025, doi:10.3390/vaccines13050495_

Round 1

Reviewer 1 Report

Comments and Suggestions for Authors

The authors tried to find conserved mutations on Usutu virus and other flaviviruses that could be used for attenuated vaccine development. It’s a smart way although live attenuated vaccines for flaviviruses always had point mutations on different genes in the whole genome. Still, there are some concerns as follows.

  1. Some fonts were not the same regarding to the cell lines.
  2. Line 221, no details were given about the generation of rUSUV-WT, E-E138K or E-306K in Materials and Methods, rather, they were shown in part 3.1 in Results section.
  3. Line 391, 20ug/ml should be 20μg/ml. So did in Figure 2B.
  4. There are some letters in lower case should be capitalized, eg. Line 422, CCl-81 should be CCL-81, Figure 4a should be Figure 4A. Please check the whole draft.
  5. For Table 4, it should be a three lines table rather than a full frame table.
  6. The author used Ifnar-/- mice as a model to evaluate USUV vaccine attenuation, it ruled out the potential of immunodeficiency animals. There should be immunocompetent controls or previous studies for the reason of using Ifnar-/- mice.
  7. The author found some reversions after infection, was it only happened on Ifnar-/- mice?
  8. Dead mice had higher NAb titres, was it because of ADE effect? So what’s the cut-off value of the protective NAb titres?

Reviewer 2 Report

Comments and Suggestions for Authors

1, Introduction would be abbreviated in a certain to make it more concise. 

2, Plaque assay should be briefly described or add a refference. 

3, in vivo and in vitro should be italic througout manuscript. 

Reviewer 3 Report

Comments and Suggestions for Authors

The study describes the search for and validation in cell and mouse models of attenuating mutations in Usutu virus to understand the virulence determinants and to develop live attenuated vaccines. First, the authors evaluated the stability of the mutations in the several passages of cell culture and investigated viral phenotypes induced by the mutations in different cellular models. Those mutations that caused attenuated phenotypes in vitro, were further tested in an IFNAR-/- mouse model. The authors found that mutations in the envelope protein glycosylation and glycosaminoglycan binding sites attenuated Usutu virus, a phenomenon observed for the other flaviviruses. Mutations introduced into the genes of the non-structural proteins either did not provide expected levels of attenuation or any phenotypic changes. Of note, infection of mice with attenuated mutant viruses resulted in high neutralizing antibody titers, as with wt virus.

The study provides a thorough analysis and detailed discussion of the results of several attenuating viral mutations, which is important for the determination of the conservative attenuating effect of mutations in the different flaviviruses, including emerging ones.  

I have a few comments and suggestion for the authors:

  • Please add a transcript of the abbreviations CPE and HEP where they first appear.
  • Line 352 – link to the wrong paragraph (should be 2.2)
  • 2 Section – Please add the link to Figure S2
  • Figure 2:

- (A) - Please indicate which cells were used to quantify relative plaque size (in the Figure caption).

- (C), (D) Please, explain why different assays (TCID50 and plaque assay) were used for the different cell lines? (Same for figure 3).   

- Could you explain the difference in the results for the USUV E-E306K mutation in TCID50 assay and plaque assay?

  • Lines 422-424 – how could you explain the different effects of NS2A mutation on replication observed for Vero and A549 cells?
  • Lines 462, 463 – incorrect links to Figure. And please, add more links to figure 6 (D, E) in the section 3.4.
  • 8B – It is unclear between which values the statistically significant differences are reported. Please, add a link to the Figure in the text.
  • Page 19, line 64 – incorrect link to Figure.
  • Figure 10: It is unclear, at which days after infection the Nabs titers were measured for each of the mouse group. If the titers were measured at different time points, then they cannot be directly compared and should be presented in a different graphical manner, not on the same scale.    
  • I would recommend that the authors present the conclusions in a tabular (or probably figure) format, summarizing the results obtained and concluding the characteristics of attenuating mutations, that are most important for clinical application in the vaccine design. Comments on the risks of introducing these mutations may also be included. This will improve the overall impression of the paper and increase the understanding of the relevance of the data for the other flaviviruses.
